# Provincial-Level CO$_2$ Emissions Intensity Inequality in China: Regional Source and Explanatory Factors of Interregional and Intraregional Inequalities

**Wanbei Jiang [1,2] and Weidong Liu [1,2,*]**

[1]   Institute of Geographic Sciences and Natural Resources Research, Chinese Academy of Sciences, Beijing 100101, China; jiangyb@igsnrr.ac.cn
[2]   Key Laboratory of Regional Sustainable Development Modeling, Chinese Academy of Sciences, Beijing 100101, China
*   Correspondence: liuwd@igsnrr.ac.cn

**Abstract:** As the largest emitter in the world, China has pledged to reduce CO$_2$ emissions intensity (CO$_2$ emissions per unit of output) by 60–65% between 2005 and 2030. CO$_2$ emissions intensity inequality analysis in China can provide a scientific basis for the Chinese government to formulate reasonable regional carbon emission abatement strategies, so as to realize the goal above. This paper adopted the Theil index to study the provincial-level CO$_2$ emissions intensity inequality in China during 2005–2015. The regional decomposition was firstly conducted and then the factors of interregional and intraregional inequalities were explored. The results show: (i) a clear increase in provincial CO$_2$ emissions intensity inequality in China has happened; (ii) this inequality and its increase were both mainly explained by the intraregional component; and (iii) the energy efficiency was the most important and positive contributor in the interregional, Eastern, Central, and Western China inequalities. Energy efficiency was also the key factor that caused the growth in interregional and Western China inequalities. However, most of the Eastern and Central China inequality increments over the whole period were respectively driven by the expanding carbonization gap and the changing GDP share, instead of the trajectory of energy efficiency. According to the results, regional emission mitigation strategies were proposed.

**Keywords:** China; CO$_2$ emissions intensity; inequality; regional decomposition; factors

## 1. Introduction

With the rapid development of the world economy, the consumption of fossil fuels has increased sharply, which has led to the release of a large amount of pollutants and greenhouse gases into the air. The global warming caused by the rapid increase of CO$_2$ (carbon dioxide) has received worldwide attention [1]. Moreover, much of the increase in emissions in the last decades can be attributed to the scale effect associated with economic growth. In this sense, and if measures to limit economic growth are not on the agenda, the reduction of global emissions necessarily requires a significant decrease in CO$_2$ emission intensities (CO$_2$ emissions per unit of output). The target of intensity decrease can also be seen as a preliminary goal to achieve the ultimate target in terms of absolute reductions [2].

As the largest emitter in the world, China plays a critical role in controlling and reducing CO$_2$ worldwide [3]. At the same time, China is still a developing country [4]. In response to global climate change and to the requirement of economic development, the Chinese government has pledged to reduce CO$_2$ emissions intensity, specifying that emissions intensity will be reduced by 40–45% by 2020 and 60–65% by 2030 compared with 2005 levels [5].

$CO_2$ emissions reduction mainly consists of optimizing interregional emission differences [6]. With diversified resource endowments (e.g., natural resources, labor, capital, and technology), there are large differences in provincial-level $CO_2$ emissions intensity in China. Existing research results show that the overall distribution pattern of provincial emissions intensity in China was low in the south and east, high in the north and west [7]. In 1997, the value of the province with the largest emissions intensity was 8.33 times that of the province with the smallest, and the gap was expanded to 8.37 times in 2010 [8].

During the period of China's seventh five-year plan (1986–1990), the country was divided into three major economic zones (i.e. Eastern, Central, and Western regions) mainly according to the geographical location, and economic construction condition. Generally, since reform and opening, the Eastern region of China has been wealthier than the Central and Western regions because of the government's early investment policies favoring coastal areas [9,10]. Provinces in the Eastern region averaged 2.7 times and 1.8 times the per capita GDP of those in the Western region in 2005 and 2015, respectively [11,12]. Using a regional decomposition of the Theil index, Clarke-Sather et al. [13] have identified the provincial-level inequality in per capita GDP within China comes primarily from the differences between the Eastern, Central, and Western regions. So, it's interesting to ask whether the greater differences in provincial-level $CO_2$ emissions intensity in China are also centered on the differences between these three regions.

This paper took 30 provinces (excluding Tibet, Hong Kong, Macao, and Taiwan) in China as research objects, and adopted the Theil index to study the provincial-level $CO_2$ emissions intensity inequality in China during 2005–2015. First, the regional decomposition was conducted to clarify the contributions of interregional and intraregional differences to the total interprovincial inequality within China, in order to help the government, determine the geographical scale of potential emissions reductions. Second, the driving factors of the interregional, Eastern, Central, and Western inequalities were analyzed to help formulate specific emissions reduction measures at different geographical scales. Therefore, the inequality analysis in this study may provide a scientific basis for the Chinese government to make reasonable regional emissions reduction strategies, so as to realize the goal that it promised to the world.

## 2. Literature Review

Beginning with Heil and Wodon [14,15], the study of the $CO_2$ emissions distribution has received much attention, especially in recent years. Scholars have employed different measures of inequality, including the Gini coefficient [16], Theil index [17], Atkinson index [18], variance [19], variation coefficient [20], density function [21], and convergence theory [22] to describe the $CO_2$ distribution. They further analyzed the source of the $CO_2$ inequality from the perspectives of groups (divided by geography, economy, etc.) [13,14,23], energy types [2,24], and economic sectors [25,26], and discussed the driving factors that caused the inequality [27–29].

Associated with inequality research on $CO_2$ emissions intensity, Camarero et al. [22] applied the Phillips and Sul [30] methodology to study convergence in $CO_2$ emissions intensity among OECD countries over the period of 1960–2008. The results highlighted that differences in emission intensity convergence are more determined by differences in the convergence of the carbonization index rather than by differences in the dynamic convergence of energy intensity. Duro et al. [2] analyzed the international inequalities in $CO_2$ emissions intensity for the period 1971–2009 and conducted group, additive, and multiplicative decompositions of the inequalities. They found the bulk of inequality between countries was explained by differences between the groups of countries considered and differences in coal and in energy intensity.

In terms of research on $CO_2$ emissions intensity inequality in China, Zhao et al. [31], Yang et al. [32], and Sun et al. [33] adopted the Theil index to depict the evolution feature of provincial $CO_2$ emissions intensity inequality, and divided China into four and eight geographical regions, and then measured the contributions of the interregional and intraregional components to provincial inequality. Li and

Jiang [34] investigated the inequalities in carbon intensity in China from 1995 to 2013, with the use of the Gini coefficient and its subgroup decomposition method. Yan et al. [7] and Wang and Yang [8] utilized the regression-based inequality decomposition method to discuss the ability of economic factors (e.g., GDP per capita, energy mix, sector composition, and urbanization level) to explain the differences in $CO_2$ emissions intensity. Zhou and Wang [35] took the ratios of the $CO_2$ emissions intensities of Western provinces to the average intensity of the Eastern provinces as the dependent variable and the ratios in economic factors as the independent variables to explore the causal factors of emissions intensity inequality.

Research on China's $CO_2$ emissions intensity inequality and its driving factors have achieved certain results. For example, scholars have reached a consistent conclusion that the $CO_2$ emissions intensity inequality at the provincial level increased over the period of 1995–2014. Through regional decomposition, most of them recognized that the inequality mainly lay within the regions. For instance, Li and Jiang [34] concluded intraregional difference was larger than the gap between Eastern, Central, and Western regions. However, there is also research that found interregional inequality was larger than the intraregional inequality, based on the four regional divisions of Northeastern, Eastern, Central, Western regions [33]. In terms of the explanatory factors, GDP per capita was identified as the most important contributor to the emissions intensity inequality. In addition, the energy mix and sector composition played significant roles as well [7,8].

In summary, previous studies have provided abundant and meaningful information for understanding the distribution of $CO_2$ emissions intensity. Although a number of studies have been reported, research gaps still exist. For example, existing factor analysis based on the Kaya identity only considered two factors, energy intensity and carbonization [2], so that we could not understand whether the contribution of energy intensity to $CO_2$ emissions intensity inequality was caused by sector composition or energy efficiency. In addition, there are few studies on explanatory factors of interregional and intraregional inequalities in $CO_2$ emissions intensity in China; most literature merely discussed the factors of total interprovincial inequality within China. In view of this, this paper firstly implemented a regional composition of provincial-level $CO_2$ emissions intensity inequality in China and then explored the factors of interregional, Eastern, Central, and Western inequalities based on the Kaya identity and logarithmic mean Divisia index (LMDI) decomposition method. This paper hopefully can contribute to the literature in the following three ways: (i) making up for the gap in the research field of explanatory factors of $CO_2$ emissions intensity inequality in China as mentioned above; (ii) separating sector composition and energy efficiency in energy intensity and considering sector composition, energy efficiency, and carbonization as three separate factors, so as to clarify the roles that sector composition and energy efficiency played in the $CO_2$ emissions intensity inequality; and (iii) verifying the regional decomposition results to help formulate a consistent conclusion.

## 3. Materials and Methods

### 3.1. Measuring $CO_2$ Emissions Intensity Inequality

Since the Theil index satisfies the axiomatic properties of an ideal inequality index including scale independence, population independence, and the principle of transfers [36] and has more advantage in the capacity to decompose, this paper adopted this index to measure the inequality in $CO_2$ emissions intensity in China. The formulation is expressed as:

$$T_C = \sum_i p_i ln\left(\frac{CI_\mu}{CI_i}\right),\tag{1}$$

where $T$ represents the Theil index; $T_C$ denotes the provincial-level inequality in $CO_2$ emissions intensity in China according to the Theil index; $CI_\mu$ is the national emissions intensity; $CI_i$ is the $CO_2$ emissions intensity of province $i$; $p_i$ is the GDP share of each province.

### 3.2. Decomposing CO₂ Emissions Intensity Inequality by Regions

The Theil index is the most attractive of all the indexes in terms of decomposition [37], especially the group decomposition. The decomposition can be expressed as follows:

$$T_C = \sum\nolimits_g p_g ln\left(\frac{CI_\mu}{CI_g}\right) + \sum\nolimits_g p_g T_g \tag{2}$$

where $CI_g$ represents the $CO_2$ emissions intensity of region $g$ (i.e. Eastern, Central, or Western region); $p_g$ is the GDP share of region $g$; $T_g$ denotes the internal inequality in region $g$. $\sum_g p_g ln\left(\frac{CI_\mu}{CI_g}\right)$ and $\sum_g p_g T_g$ are respectively the interregional ($T_{inter}$) and intraregional ($T_{intra}$) components of the national inequality ($T_C$).

### 3.3. Decomposing CO₂ Emissions Intensity Inequality by Factors

Wang and Zhou [26] applied the LMDI-I(logarithmic mean Divisia index method I) introduced by Choi and Ang [38] to assess global $CO_2$ emission inequality. This paper referred to their method to decompose the interregional, Eastern, Central, and Western inequalities in emissions intensity by three factors of the sector composition (*S*), energy efficiency (*E*) and carbonization (*C*). The inequality in $CO_2$ emissions intensity according to the Theil index can be formulated as:

$$T = T_S + T_E + T_C, \tag{3}$$

where $T_S$, $T_E$, and $T_C$ denote the contributions of the dispersions of sector composition, energy efficiency, and carbonization to the inequality in $CO_2$ emissions intensity respectively.

Suppose there is a change in $T$ ($\Delta T$) between year 0 (the base year) and year $t$. Based on the Marshall–Edgeworth model [39], $\Delta T$ can be modelled as:

$$\Delta T = \Delta T_S{}^d + \Delta T_E{}^d + \Delta T_C{}^d + \Delta T^p, \tag{4}$$

where $\Delta T_X{}^d$ denotes the $CO_2$ emissions intensity inequality variation attributed to the change in the dispersion of factor $X$ (i.e. *S*, *E*, or *C*). Equation (1) shows that a change in GDP share can also result in variation in the emissions intensity inequality. So, the item $\Delta T_p$ is used to capture the impact of changing GDP share on the inequality. A detail description of the factor decomposition is provided in Appendix A.

### 3.4. Data

The burning of fossil fuels is a major source of $CO_2$ emissions. Since China's existing statistical data did not directly provide provincial or sectoral $CO_2$ emissions data, prior studies generally used fossil energy consumption to measure $CO_2$ emissions. In this paper, sectoral $CO_2$ emissions in each province were estimated according to the methods provided by IPCC (2006) [40]. The specific estimation formula is as follows:

$$CO_{2ij} = \frac{44}{12} \times \sum\nolimits_k AE_k \times LHV_k \times CC_k \times CO_k, \tag{5}$$

where $k$ represent kth type fossil energy; $CO_{2ij}$ denotes energy-burned $CO_2$ emissions of sector $j$ in province $i$; $AE_k$, $LHV_k$ $CC_k$, and $CO_k$ respectively denote consumption, average low heat value, carbon content, and carbon oxidation factor of fossil energy $k$.

The fossil energy consumption data (AE) of 30 provinces (excluding Tibet, Hong Kong, Macao, and Taiwan) in China used in this paper are from the Energy Balance Table by Region in China Energy Statistical Yearbook (2006–2016) [41]. The categories of energy considered include 17 (years 2005–2009) and 27 (years 2010–2015) fossil fuels in the Energy Balance Table by Region. Average low heat value (LHV) comes from the Conversion Factor from Physical Units to Coal Equivalent in China Energy

Statistical Yearbook. Carbon content (CC) and carbon oxidation factor (CO) are sourced from the Guidelines for Provincial Greenhouse Gas Inventories (2011) [42] and IPCC Guidelines for National Greenhouse Gas Inventories (2006) [40].

Referring to the division of economic sectors in the Energy Balance Table by Region in China Energy Statistical Yearbook, this paper divided the national economic sectors into six major sectors of agriculture, forestry, animal husbandry and fishery (S1); industry (S2); construction (S3); transport, storage and post (S4); wholesale, retail trade and hotel, restaurants (S5); and others (S6). The final energy consumption of each sector was used for estimating the sectoral $CO_2$ emission. Thermal power and heating supply in the energy transformation process also emitted $CO_2$, therefore their fossil energy amount consumption was counted in the industrial sector. In addition, energy consumption as non-energy use in the industrial sector was subtracted.

The annual data for GDP and sectoral value added at the provincial level from 2005 to 2015 were collected from the China Statistical Yearbook (NBS, 2006–2016) [12,43]. In order to obtain the real GDP and sectoral value added, this paper utilized the indexes (preceding year = 100) to revise them, and finally measured them with the constant price in 2005.

In order to analyze the regional sources of provincial $CO_2$ emissions intensity inequality in China, the provinces to be included in the Eastern, Central, and Western regions were determined. According to the official Chinese government regional classifications, the Eastern region includes Beijing, Tianjin, Hebei, Shanghai, Jiangsu, Zhejiang, Fujian, Shandong, Guangdong, Hainan, and Liaoning; the Central region covers eight provinces of Jilin, Heilongjiang, Shanxi, Anhui, Jiangxi, Henan, Hubei, and Hunan; the Western region includes Inner Mongolia, Guangxi, Chongqing, Sichuan, Guizhou, Yunnan, Shaanxi, Gansu, Qinghai, Ningxia, and Xinjiang.

## 4. Results

### 4.1. Spatial and Temporal Distribution of $CO_2$ Emissions Intensity in China

In 2005 and 2015, it can be seen from Figure 1 that the $CO_2$ emissions intensity levels of the Central and Western exceeded, while those of the Eastern region were slightly below, the national levels. At the provincial level, the Eastern provinces of Beijing, Guangdong, and Shanghai have the lowest $CO_2$ emissions intensities, while the Western provinces of Ningxia and Inner Mongolia and the Central province of Shanxi have relatively high emissions intensities. In terms of the trend, national $CO_2$ emissions intensities decreased from a value of 0.26 (kg per yuan of output) in 2005 to 0.17 in 2015, the minimum level of the time series (2005–2015). The reduction was significant in all regions and almost in all the provinces except Xinjiang and Ningxia. This indicated that the increase in total $CO_2$ emissions was lower than the growth in GDP over the period studied at the national, regional, and provincial levels. However, the decreasing rates were rather heterogeneous for different regions and provinces. For the Eastern and Central regions, the decreasing rates were 39% and 40% respectively, which were higher than that of the Western region (30%). At the provincial level, Beijing in the East and Qinghai in the West respectively had the fastest (61%) and slowest (13%) rates of decline. Furthermore, Xinjiang and Ningxia owned the growth rates of 26% and 2%. This may indicate that provinces in the Eastern and Central regions paid more attention to energy conservation and low-carbon development and emitted less additional $CO_2$ than the Western provinces in the process of economic development.

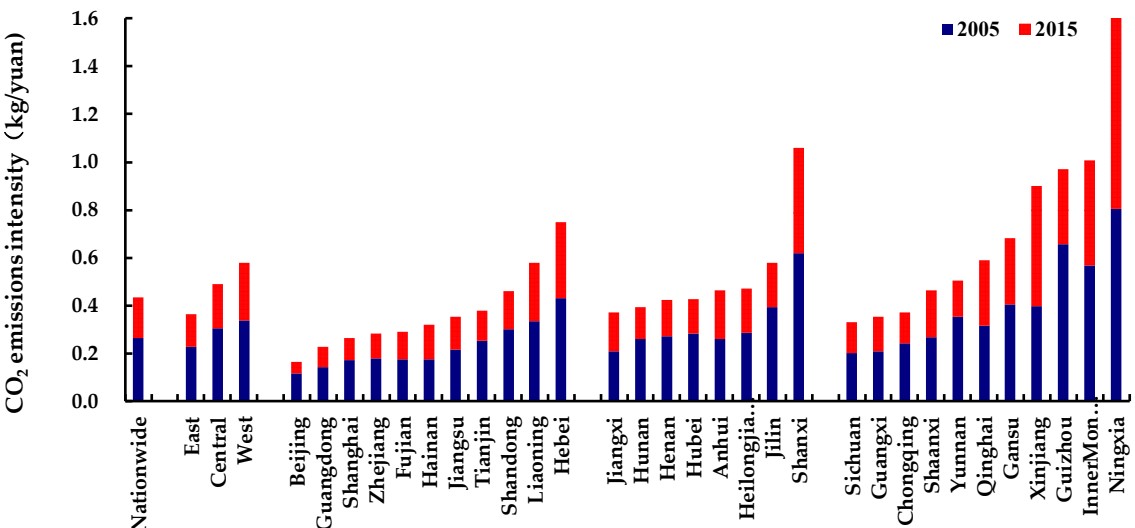

**Figure 1.** $CO_2$ emissions intensity of China, three regions, and provinces in 2005 and 2015. Notes: provinces are arranged by regions.

## 4.2. $CO_2$ Emissions Intensity Inequality Measurement according to the Theil Index and the Regional Decomposition of It

Table 1 shows provincial inequalities in $CO_2$ emissions intensity according to the Theil index (first column) increased continuously over the period: from a value of 0.083 in 2005 to 0.125 in 2015. Sudden increases happened in 2008, 2011, and 2013. The result implied that the disparities in $CO_2$ emissions intensity among the 30 provinces expanded during 2005–2015. The reason may be the values of provinces with lower emissions intensity (Beijing, Shanghai, Guangdong, etc.) have further decreased, while the values of provinces with higher emissions intensity (Ningxia, etc.) have increased slightly (Figure 1). The finding is consistent with the conclusions reported in Yan et al. [7] according to the Theil index, Gini coefficient, and mean logarithmic deviation, and coincided with the results based on convergence analysis for $CO_2$ emissions intensity [35].

**Table 1.** Provincial-level inequality in $CO_2$ emissions intensity in China according to the Theil index decomposed by regions, 2005–2015.

| | $CO_2$ Emissions Intensity Inequality | Interregional Component | Intraregional Component |
|---|---|---|---|
| 2005 | 0.083 | 0.015 (18%) | 0.068 (82%) |
| 2006 | 0.087 | 0.020 (23%) | 0.067 (77%) |
| 2007 | 0.091 | 0.024 (26%) | 0.067 (74%) |
| 2008 | 0.098 | 0.028 (28%) | 0.07 (72%) |
| 2009 | 0.101 | 0.030 (30%) | 0.071 (70%) |
| 2010 | 0.103 | 0.025 (24%) | 0.078 (76%) |
| 2011 | 0.112 | 0.029 (26%) | 0.083 (74%) |
| 2012 | 0.116 | 0.031 (26%) | 0.086 (74%) |
| 2013 | 0.123 | 0.027 (22%) | 0.096 (78%) |
| 2014 | 0.125 | 0.027 (22%) | 0.097 (78%) |
| 2015 | 0.125 | 0.024 (19%) | 0.101 (81%) |

Notes: within brackets the relative weight of each component on the national inequality in emissions intensity.

A regional decomposition analysis of the Theil index revealed that the intraregional component contributions to $CO_2$ emissions intensity inequalities in China were consistently much larger than the interregional component. In the period from 2005 to 2015, the intraregional component explained between 70% and 82% of national inequalities (Table 1). This result was consistent with the regional decomposition result of Yue et al. [44] which also divided China into the Eastern, Central, and Western

regions, although the overlapping study period was only from 2005 to 2007. However, it was opposite to the results of Duro et al. [2] which demonstrated the regional groups defined according to geographical–economic criteria appeared to be good proxies of the international differences in $CO_2$ emissions intensity.

The increase in national inequalities in $CO_2$ emissions intensity in China was mainly explained by the increase in inequalities within the regions of provinces. Although there was also an increase in interregional inequalities over the period, an increment of 0.009 was much less than that experienced by inequalities within the regions (0.033) (Table 1).

The intraregional component of the national inequality could be further decomposed into contributions of regions. The internal inequality of each region, appropriately weighted by the GDP share, produced the intraregional component of the national inequality. Table 2 shows that the Eastern region was the largest contributor to intraregional inequality. This may attribute to its higher GDP share of 57%–60%. The Western region acted as the second-largest contributor and contributed 20%–28% to internal inequalities of all regions. With a GDP share of 23%–24%, the Central region only contributed 12%–15% to intraregional inequality. Table 2 shows that the absolute contributions of three regions all experienced an increase. As a result of a relatively large increase in internal inequality and larger GDP share, the Eastern region contributed more than the other two regions to the growth in intraregional inequality.

**Table 2.** Details of intraregional component of provincial-level inequality in $CO_2$ emissions intensity in China according to the Theil index, 2005–2015.

| | East | | | Central | | | West | | |
|---|---|---|---|---|---|---|---|---|---|
| | Internal Theil (Tg) | GDP % (pg) | Absolute Contribution | Internal Theil (Tg) | GDP %(pg) | Absolute Contribution | Internal Theil (Tg) | GDP % (pg) | Absolute Contribution |
| 2005 | 0.071 | 60% | 0.042 | 0.043 | 23% | 0.010 | 0.091 | 17% | 0.015 |
| 2006 | 0.067 | 60% | 0.040 | 0.040 | 23% | 0.009 | 0.105 | 17% | 0.018 |
| 2007 | 0.068 | 60% | 0.041 | 0.035 | 23% | 0.008 | 0.105 | 17% | 0.018 |
| 2008 | 0.071 | 60% | 0.042 | 0.045 | 23% | 0.010 | 0.103 | 17% | 0.017 |
| 2009 | 0.073 | 59% | 0.043 | 0.046 | 23% | 0.011 | 0.099 | 17% | 0.017 |
| 2010 | 0.087 | 59% | 0.051 | 0.046 | 24% | 0.011 | 0.092 | 17% | 0.016 |
| 2011 | 0.086 | 58% | 0.050 | 0.045 | 24% | 0.011 | 0.126 | 18% | 0.022 |
| 2012 | 0.088 | 58% | 0.051 | 0.049 | 24% | 0.012 | 0.127 | 18% | 0.023 |
| 2013 | 0.103 | 58% | 0.060 | 0.057 | 24% | 0.014 | 0.125 | 18% | 0.023 |
| 2014 | 0.102 | 58% | 0.059 | 0.058 | 24% | 0.014 | 0.134 | 18% | 0.025 |
| 2015 | 0.105 | 57% | 0.060 | 0.053 | 24% | 0.013 | 0.153 | 19% | 0.028 |

Notes: absolute contribution was calculated as the product of the internal inequality of each region (according to the Theil index) and its corresponding GDP share.

The region with the greatest internal inequality in $CO_2$ emissions intensity was the Western region, followed by the Eastern region. The Central region had the smallest internal inequality, which explained why it owned the lowest contribution described above. Taking the Eastern and Western regions of China respectively as the sub-national analogs of developed and developing nations, this result complements those found in Aldy [45] and Duro et al. [2] that question the convergence in developing countries in terms of $CO_2$ emissions: developing countries have higher internal inequality in emissions intensity.

*4.3. Factor Decomposition of Interregional and Intraregional Inequalities in $CO_2$ Emissions Intensity in China*

In this part, we conducted a factor decomposition in order to clarify the roles of the sectoral composition, energy efficiency, and carbonization in the observed interregional, Eastern, Central, and Western inequality patterns in $CO_2$ emissions intensity. For interregional inequality, energy efficiency disparity among regions contributed the most (contribution rate of 57%–104%). Notably, it contributed more than 80% since 2008. Much lower was the contribution of the sectoral composition (contribution rate of 4%–37%). While it had a contribution rate of 37% in 2005, it lost its explanatory capacity after the first year of the period. The carbonization contributed the least to the interregional

inequality during 2005–2009. In fact, its contribution to the interregional component became negative at last and this inequality tended to compensate for the inequality from the sectoral composition disparity (Figure 2a).

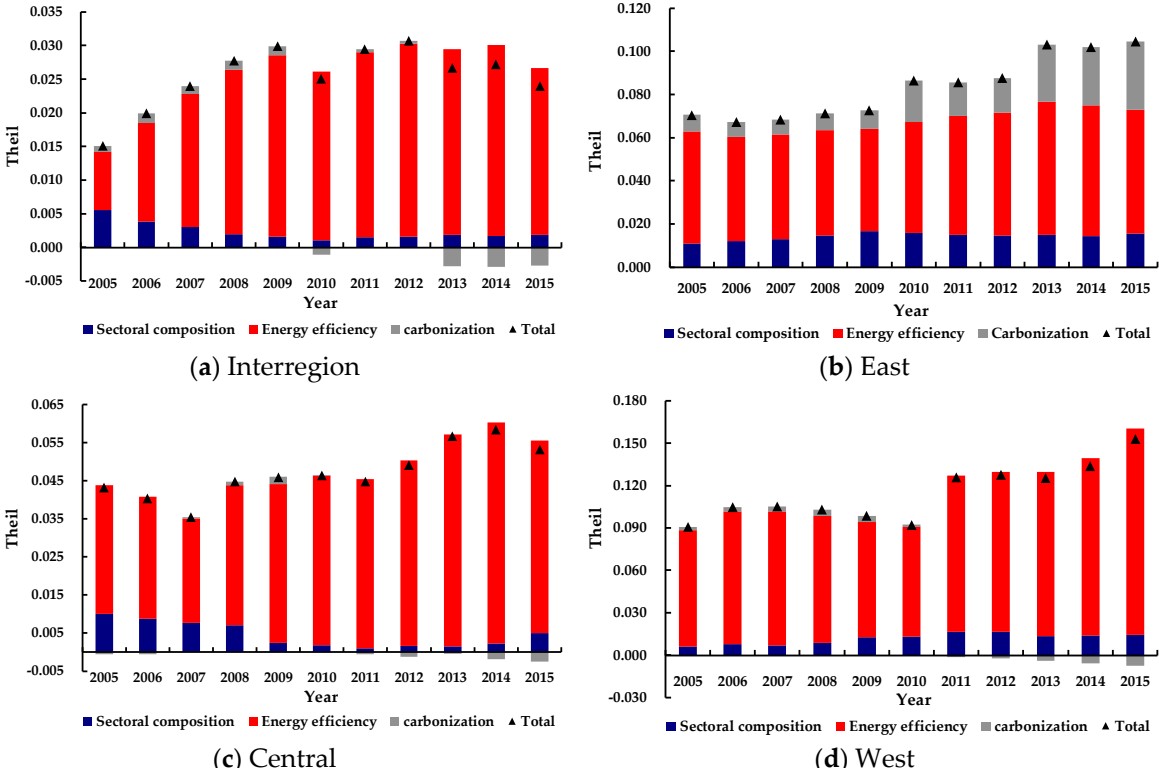

**Figure 2.** Interregional and intraregional inequalities in $CO_2$ emissions intensity in China decomposed by factors, 2005–2015.

With respect to the Eastern region, all the factors made net positive contributions to the internal inequality. The main explanatory factor was associated with the energy efficiency, with a contribution of between 55% and 74% over the whole period. For the sectoral composition and carbonization factors, the former contributed a little more (16%−23%) than the later (10%−12%) in 2005–2009, but the opposite was true in 2010–2015 (contribution rates of 14%−18% and 18%−30% for the sectoral composition and carbonization, respectively). This observed phenomenon was mainly attributed to the greater inequality growth of the carbonization compared to the sectoral composition (Figure 2b).

The Central region was very similar to the interregional component in terms of the contributions of the three factors to the inequality in $CO_2$ emissions intensity. The energy efficiency was also the largest contributor, whose contribution rate was 77%−99% over the period considered. The sectoral composition had a contribution of 23% in 2005 and 9% in 2015, which was also a positive contributor to the inequality and lost its explanatory capacity at last. The carbonization contributed negatively but slightly in most years, except in 2007–2010 (Figure 2c).

For the Western region, the inequality in $CO_2$ emissions intensity was predominantly explained by the energy efficiency as well, with a contribution of 83%−96% over the whole period. The inequality from the sectoral composition disparity was positive. It increased during 2005–2011, and then decreased slightly in the last years of the period. The carbonization was the least important contributor to the Western inequality. Its impact was positive during 2005–2010, and became increasingly negative from 2011 (Figure 2d).

Next, we focused on the evolutions of the interregional, Eastern, Central, and Western inequalities in $CO_2$ emissions intensity and the effects of factors considered on them. Since the evolution trend of $CO_2$ emissions intensity inequality fluctuated during the period studied (as shown in Figure 2),

we further divided the entire period into two sub-periods, i.e., period I (2005–2010) and period II (2010–2015).

According to the Theil index, an increase of 0.010, −0.001, and 0.009 occurred in the interregional $CO_2$ emissions intensity inequality in period I, period II, and the whole period, respectively. The inequality increased in the first sub-period, and the trend was reversed in the latter sub-period. Decomposing the evolution of inequality by factors uncovered that the expanding disparity in energy efficiency levels among regions was the only factor that enhanced the interregional inequality in period I. The positive effect was stronger (with a $\Delta T_E^d$ of 0.020) than the sum of the negative effects from the other three factors. So, even though narrowing differences in the sectoral composition and carbonization, and the changing GDP share all helped reduce the inequality, the positive contribution of the energy intensity to the interregional inequality increase could not be offset. In period II, the enhancing effect of the energy efficiency was weakened (with a $\Delta T_E^d$ of 0.008), and the negative effect of the changing GDP share was further increasing. Thus, the interregional inequality in $CO_2$ emissions intensity decreased a little. Over the whole period studied (2005–2015), the energy efficiency enhanced the interregional inequality, and the other three factors all contributed to decreasing the inequality (Table 3).

**Table 3.** Changes of interregional and intraregional $CO_2$ emissions intensity inequalities in China decomposed by factors, 2005–2015.

|  | $\Delta T_S^d$ | $\Delta T_E^d$ | $\Delta T_C^d$ | $\Delta T^p$ | $\Delta T$ |
|---|---|---|---|---|---|
| **2005/2010** | | | | | |
| Interregion | −0.005 | 0.020 | −0.002 | −0.003 | 0.010 |
| East | 0.008 | 0.002 | 0.012 | −0.006 | 0.016 |
| Central | −0.011 | 0.008 | 0.001 | 0.006 | 0.003 |
| West | 0.011 | −0.006 | 0.000 | −0.003 | 0.001 |
| **2010/2015** | | | | | |
| Interregion | 0.000 | 0.008 | −0.002 | −0.008 | −0.001 |
| East | 0.003 | 0.005 | 0.013 | −0.003 | 0.018 |
| Central | 0.004 | −0.002 | −0.002 | 0.007 | 0.007 |
| West | 0.001 | 0.065 | −0.009 | 0.005 | 0.061 |
| **2005/2015** | | | | | |
| Interregion | −0.006 | 0.028 | −0.004 | −0.010 | 0.009 |
| East | 0.011 | 0.008 | 0.025 | −0.010 | 0.034 |
| Central | −0.007 | 0.002 | −0.001 | 0.015 | 0.010 |
| West | 0.013 | 0.058 | −0.010 | 0.001 | 0.062 |

Notes: for the interregion, East, Central, or West, $\Delta T$, $\Delta T_S^d$, $\Delta T_E^d$, $\Delta T_C^d$, and $\Delta T^p$ respectively denote the total inequality change according to the Theil index, and the components caused by sector composition dispersion change, energy efficiency dispersion change, carbonization dispersion change, and GDP share change.

With respect to the Eastern inequality in $CO_2$ emissions intensity, the increments of the Theil index were 0.016, 0.018, and 0.034, respectively, for period I, period II, and the whole period. The added values of period I and period II were nearly equal. The decomposition of the inequality evolution indicated that the changing GDP share was the only factor that helped reduce the inequality, the other three factors all caused the enhancement of the inequality, whatever the period considered. Among them, the carbonization played the most important role, with contributions of 0.012, 0.013, and 0.025 in period I, period II, and the whole period. The sectoral composition contributed less than the carbonization, but more than the energy efficiency in period I. However, it lost a certain importance in period II so that it became the least significant factor in the inequality growth. At the same time, the energy efficiency exerted more and more important effect from period I to II. For the whole period (2005–2015), the sectoral composition was still the second-largest contributor to the Eastern equality growth (Table 3).

In the Central region, the Theil index associated with the $CO_2$ emissions intensity respectively increased an amount of 0.003, 0.007, and 0.010 in period I, period II, and the whole period studied. In period I, the sectoral composition disparity among the Central provinces contracted, so it helped

to reduce the Central inequality. On the other hand, the differences in the energy efficiency and the carbonization expanded, thus they facilitated the inequality increase. In period II, the above three factors all changed their roles in the inequality variation. The effect of the changing GDP share was invariantly positive, no matter in period I or II. On the whole, the change of GDP share determined the growth in the emissions intensity inequality. If kept constant, the Central inequality might have decreased. The energy efficiency also contributed to the inequality increase, but to a relatively small extent. The narrowing differences among Central provinces in the other two factors were helpful for reducing the emissions intensity inequality (Table 3).

According to our results, the Western inequality mainly increased in period II. The Theil index increased by 0.062 over the whole period, of which 0.001 and 0.061 were in period I and II, respectively. It seems that the Western inequality in period I almost kept constant. This is because the negative effects of the narrowing energy efficiency and the changing GDP share nearly compensated for the positive effect of sectoral composition on the inequality growth. The influence of the carbonization was almost negligible. In period II, the difference in the sectoral composition was still expanding, to a much lower degree compared to period I. The energy efficiency and the GDP share turned to widen the emissions intensity gap. The former became the most important and dominant promoter in the inequality, causing an increased Theil index of 0.065. In this period, the carbonization promoted the equality singly. For the whole period, the energy efficiency played the most important role in enhancing the Western inequality. In addition, the sectoral structure and the GDP share also exerted positive effects, while carbonization was the driver in the opposite direction (Table 3).

## 5. Discussion and Policy Implications

The regional decomposition results demonstrated that throughout the decade studied, the contribution of the intraregional component to the national inequality in $CO_2$ emissions intensity was larger than that of the interregional component. Within the individual Eastern, Central, and Western regions, the inequalities were all much greater than those between the regions. This illustrated that the $CO_2$ emissions intensity inequality was not primarily regional in nature. Although the economic levels of the Eastern, Central, and Western regions are quite different, the difference in $CO_2$ emissions intensity is not obvious. Provinces with similar economic development levels and geographical position achieved very different $CO_2$ emissions intensities.

The factor decomposition results indicated that no matter whether for the interregional, Eastern, Central, or Western inequalities in $CO_2$ emissions intensity, the energy efficiency was the most important and positive contributor. This result was in accord with Barry Commoner's research conclusions, which also recognized the significant role that production technology played in environmental quality, regardless of whether they were in developed or developing countries [46,47]. We calculated the variation coefficient of industrial energy efficiency in 2015 as an instance, and found that the variation coefficient of the three regions was 0.32, much lower than that of Eastern provinces (0.63), Central provinces (0.67), and Western provinces (0.82). This phenomenon may explain why the internal $CO_2$ emissions intensity inequalities in the Eastern, Central, and Western regions were greater than the interregional inequality.

According to the results, it could be seen that China's $CO_2$ emissions intensity reduction potential lay within the regions, especially within the Eastern and Western regions. Therefore, we mainly put forward the following political suggestions for each region based on the factor decomposition results:

First, as the bulk of the inequality in emissions intensity was still attributable to energy efficiency disparity, reductions in the Eastern inequality should involve processes of convergence towards enhanced levels of energy efficiency. As Figure 3a shows, the energy efficiency of Hainan province was much poorer than the other Eastern provinces, which was mainly due to the low efficiency of the industrial sector. In 2015, above-scale industrial energy consumption in Hainan province accounted for 57.5% of the total energy consumption of the whole society, while industrial added value only accounted for 12.12% of the total social output value. Therefore, Hainan province should improve the overall

efficiency of energy utilization from the aspects of source control of energy demand and optimization of the whole process of energy utilization. Especially in the industrial sector, it is necessary to improve the energy use efficiency of this sector by promoting the technical transformation of energy-intensive enterprises such as cement, paper, petrochemical, metal smelting, and electricity.

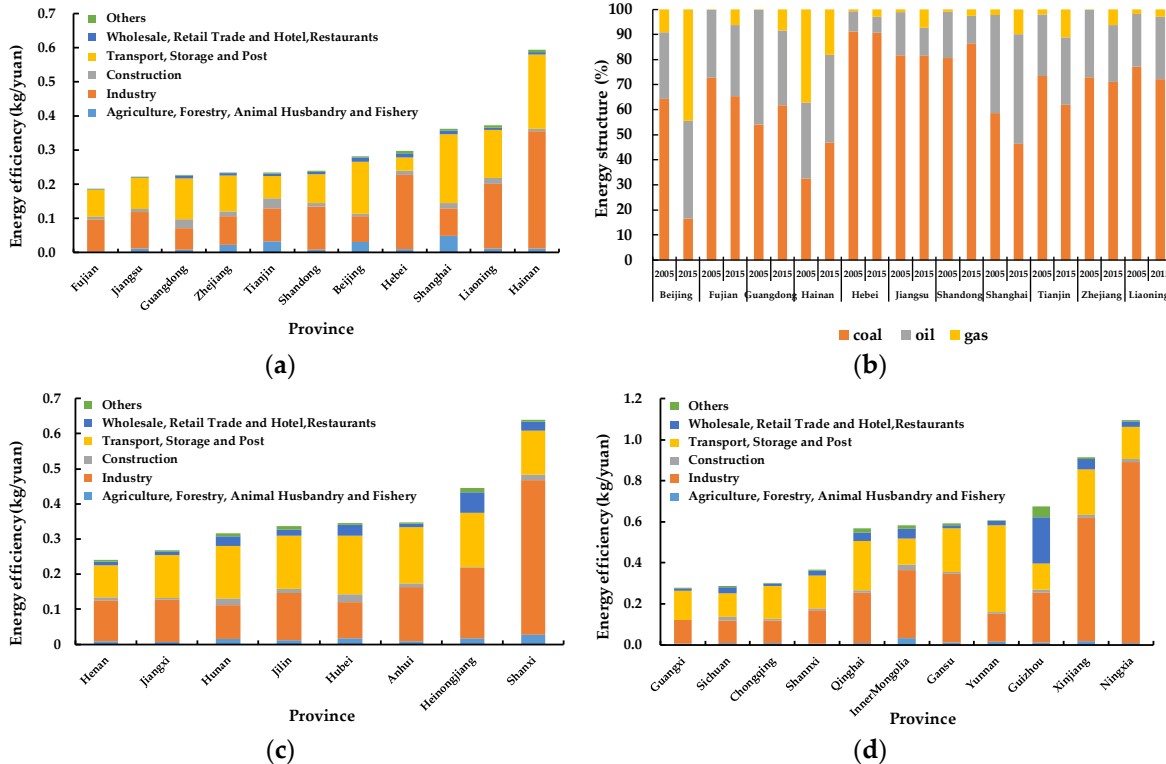

**Figure 3.** Energy efficiency values sector by sector of Eastern (**a**), Central (**c**), and Western (**d**) provinces in China in 2015 and energy structures of Eastern provinces in 2005 and 2015 (**b**).

In addition to the most important factor of the energy efficiency, we should increasingly focus on the carbonization, which exerted more and more important influence on the Eastern inequality in $CO_2$ emissions intensity. Figure 3b shows that from 2005 to 2015, the energy structures of Eastern provinces gradually became cleaner, with most provinces seeing a decrease in the proportion of coal and an increase in the proportion of natural gas. Among them, Beijing's energy restructuring achieved striking results, which might have caused the wider gap in energy structure between Eastern provinces. It gradually realized a cleaner energy system by cutting coal, bringing in foreign electricity, and increasing natural gas, which is a process that should ultimately be followed by other provinces.

Second, for the Central region, although the disparity in energy intensity did not increase too much, we still need to start with this factor to reduce the $CO_2$ emissions intensity gap because of its dominance in the intensity inequality. Among Central provinces, Shanxi province's energy efficiency was eye-catching (Figure 3c). Shanxi province is one of the largest coal-reliant provinces in China, with a total reserve of 267.4 billion tons, accounting for about a quarter of the national total. Therefore, the energy utilization level can be improved from the aspects of efficient utilization of coal and emission reduction of coal-fired power plants.

Third, for the Western region, the energy efficiency was the most promising factor to reduce the $CO_2$ emissions intensity inequality, as well. Figure 3d shows that Ningxia and Xinjiang had higher, and Guangxi, Sichuan, and Chongqing had lower energy efficiency values than other Western provinces on the whole, especially in the industrial sector. During the 12th five-year plan period, Guangxi province vigorously carried out energy-saving renovation projects of industrial boilers (kilns) transformation, cogeneration, energy saving of motor systems, optimization of the energy system,

utilization of residual heat and pressure, building energy savings, and green lighting. The clean and efficient use of coal in Sichuan province steadily improved, and the project to upgrade coal-fired power units to ultra-low emissions launched. Chongqing required key energy-using enterprises to allocate a certain amount of funds for energy conservation technology progress every year, and speed up the elimination and upgrading of outdated and energy-intensive mechanical and electrical equipment, such as old motors, transformers, fans, and pumps, and actively use the equipment promoted by the state. The energy efficient strategies and technologies utilized in Guangxi, Sichuan, Chongqing, etc. provinces could be transferred to Ningxia, Xinjiang, etc. provinces to avoid further widening the gap in energy intensity within the region.

## 6. Conclusions

In this paper, we have applied the Theil index to measure the provincial-level inequalities in $CO_2$ emissions intensity in China for the period 2005–2015, decomposed them into a part attributable to differences between regions (interregional component) and another attributable to the internal differences within these regions (intraregional component), and then analyzed the contributions of different factors to these two components and their trajectories.

The results showed that the reduction in overall $CO_2$ emission intensity happened with a clear increase in its provincial dispersion. From the regional decomposition results, it could be seen that this inequality and its increase were both mainly explained by the intraregional component. The region with the greatest internal inequality was the Western region, followed by the Eastern, and the Central regions, in turn.

From the factor composition results, we determined that the energy efficiency was the most important and positive contributor whether looking at the interregional, Eastern, Central, or Western inequality. It was also the key factor that caused the interregional and Western inequality growths. It is worth noting that most of the Eastern and Central inequality growths over the whole period were driven by the expanding carbonization gap and the changing GDP share, respectively, rather than the trajectory of energy efficiency.

**Author Contributions:** Conceptualization, W.L. and W.J.; methodology, W.J.; software, W.J.; validation, W.L.; formal analysis, W.J.; investigation, W.J.; resources, W.L.; data curation, W.J.; writing—original draft preparation, W.J.; writing—review and editing, W.L.; visualization, W.J.; supervision, W.L.; project administration, W.L.; funding acquisition, W.L. All authors have read and agreed to the published version of the manuscript.

**Funding:** This research was funded by the National Key Research and Development Program of China, grant number 2016YFA0602800.

**Conflicts of Interest:** The authors declare no conflict of interest.

## Appendix A Decomposing $CO_2$ Emissions Intensity Inequality by Factors

The interregional inequality ($T_{inter}$) was taken as an example to illustrate the decomposing process by factors. The factor decompositions of Eastern, Central, and Western inequalities were the same as that of the interregional inequality. The $CO_2$ emissions intensity of region $g$ can be formulated as follows:

$$CI_g = \frac{CO_{2g}}{GDP_g} = \sum_j \frac{GDP_{gj}}{GDP_g} \frac{EN_{gj}}{GDP_{gj}} \frac{CO_{2gj}}{EN_{gj}} = \sum_j S_{gj} E_{gj} C_{gj}, \tag{A1}$$

where $CO_{2g}$ and $GDP_g$ denotes the $CO_2$ emissions and gross regional product of region $g$ respectively, $CO_{2gj}$, $EN_{gj}$, $GDP_{gj}$ are $CO_2$ emission amount, energy consumption, and value added of sector j in region g. $S_{gj} = GDP_{gj}/GDP_g$ is the sector composition of region $g$ in terms of sectoral value added share; $E_{gj} = EN_{gj}/GDP_{gj}$, defined as the ratio of sectoral energy consumption to the value added, indicates the energy efficiency of sector $j$ in region $g$—the higher the value, the lower the efficiency level; $C_{gj} = CO_{2gj}/EN_{gj}$ is the carbonization of sector $j$ in region $g$, which characterized the energy structure. $S_{gj}$, $E_{gj}$, and $C_{gj}$ together explain the $CO_2$ emissions intensity of region $g$. The inequality in $CO_2$ emissions intensity can, therefore, be attributed to these three factors.

The logarithmic term in interregional inequality ($T_{inter}$) can be expressed as:

$$\frac{CI_\mu}{CI_g} = \frac{\sum_j S_{\mu j} E_{\mu j} C_{\mu j}}{\sum_j S_{gj} E_{gj} C_{gj}} = D_S^{\mu,g} D_E^{\mu,g} D_C^{\mu,g}, \tag{A2}$$

where $D_S^{\mu,g}$, $D_E^{\mu,g}$ and $D_C^{\mu,g}$ respectively capture the impacts of sectoral composition disparity, energy efficiency disparity, and carbonization disparity on the difference between the national level and region $g$ in $CO_2$ emissions intensity. According to LMDI-I, the impacts of these three factors can be measured as:

$$D_S^{\mu,g} = exp\left(\sum_j w_{gj} ln \frac{S_{\mu j}}{S_{gj}}\right), \tag{A3}$$

$$D_E^{\mu,g} = exp\left(\sum_j w_{gj} ln \frac{E_{\mu j}}{E_{gj}}\right), \tag{A4}$$

$$D_C^{\mu,g} = exp\left(\sum_j w_{gj} ln \frac{C_{\mu j}}{C_{gj}}\right) \tag{A5}$$

So, the interregional inequality in $CO_2$ emissions intensity can be formulated as:

$$\begin{aligned} T_{inter} &= \sum_g p_g ln D_S^{\mu,g} + \sum_g p_g ln D_E^{\mu,g} + \sum_g p_g ln D_C^{\mu,g} \\ &= \sum_{gj} p_g w_{gj} ln \frac{S_{\mu j}}{S_{gj}} + \sum_{gj} p_g w_{gj} ln \frac{E_{\mu j}}{E_{gj}} + \sum_{gj} p_g w_{gj} ln \frac{C_{\mu j}}{C_{gj}} \\ &= T_{interS} + T_{interE} + T_{interC}, \end{aligned} \tag{A6}$$

where $T_{interS}$, $T_{interE}$, and $T_{interC}$ denote the contributions of sector composition, energy efficiency, and carbonization to the interregional inequality, respectively. $w_{gj} = L(CI_{gj}, CI_{\mu j})/L(CI_g, CI_\mu)$ is the weight function; $CI_{gj} = CO_{2gj}/GDP_g$. $L(\cdot, \cdot)$ is the logarithmic mean function that is defined as:

$$L(a,b) = \begin{cases} \frac{a-b}{lna - lnb}, & if\ a \neq b \\ a, & if\ a = b \end{cases} \tag{A7}$$

Suppose there is a change in $T$ ($\Delta T$) between year 0 (the base year) and year $t$. According to Equation (A6), $\Delta T_{inter}$ can be modeled as:

$$\Delta T_{inter} = \Delta T_{interS} + \Delta T_{interE} + \Delta T_{interC} \tag{A8}$$

Equation (A6) shows that two variables of $p_g$ and $ln D_X^{\mu,g}$ are responsible for the change in $T_{interX}$. $X$ denotes these three factors considered in this paper, i.e., sector composition ($S$), energy efficiency ($E$), and carbonization ($C$). So, it may be interesting to know whether the trajectories of $T_{interX}$ ($\Delta T_{interX}$) are caused by changes in the dispersion of factor $X$, or by changes in $p_g$. Based on Marshall–Edgeworth model [39], $\Delta T_{interX}$ can be decomposed as:

$$\Delta T_{interX} = \Delta T_{interX}{}^d + \Delta T_{interX}{}^p, \tag{A9}$$

where $\Delta T_{interX}{}^d$ and $\Delta T_{interX}{}^p$, respectively, denote the impacts of changes in the dispersion of factor $X$ and in $p_g$ on $T_{interX}$. Their computational formulas are defined as:

$$\Delta T_{interX}{}^d = \frac{1}{2} \sum_{gj} \left(p_g{}^t + p_g{}^0\right)\left(w_{gj}{}^t ln \frac{X_{\mu j}{}^t}{X_{gj}{}^t} - w_{gj}{}^0 ln \frac{X_{\mu j}{}^0}{X_{gj}{}^0}\right), \tag{A10}$$

$$\Delta T_{interX}{}^p = \frac{1}{2} \sum_{gj} \left(p_g{}^t - p_g{}^0\right)\left(w_{gj}{}^t ln \frac{X_{\mu j}{}^t}{X_{gj}{}^t} + w_{gj}{}^0 ln \frac{X_{\mu j}{}^0}{X_{gj}{}^0}\right). \tag{A11}$$

So, Equation (A9) can be rewritten as:

$$\Delta T_{inter} = \Delta T_{interS}{}^{d} + \Delta T_{interE}{}^{d} + \Delta T_{interC}{}^{d} + \Delta T_{inter}{}^{p},$$ (A12)

where $\Delta T_{inter}{}^{p} = \sum_{X} \Delta T_{interX}{}^{p}$ is the sum of the effect of changing GDP share ($p_g$).

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
