# Peer review of "Provincial-Level CO2 Emissions Intensity Inequality in China: Regional Source and Explanatory Factors of Interregional and Intraregional Inequalities"

_sustainability, doi:10.3390/su12062529_

Round 1
Reviewer 1 Report
Some Recommendations:
-Read and include a brief discussion of this papers by Barry Commoner:
-B. Commoner, «Population, development, and the environment: trends and key issues in the developed countries», International Journal of Health Services, vol. 23, n. 3, 1993, pp. 519-539.
-B. Commoner, «Rapid population growth and environmental stress», International Journal of Health Services, vol. 21, n. 2, 1991, pp. 199-227.
Reviewer 2 Report
As a reviewer I have the following remarks
- The paper is well written and presented.
- Line 31, in the first use of CO2, I suggest to add the term “(carbon dioxide)”.
- After your formulas, should be “where” not “Where”.
- Line 201, is it a special reason to have 44/12 rather than 11/3?
- Fig 2 (a). Is it possible to put x-labels down, to don’t overlay the bars.
Thank you.
Reviewer 3 Report
The authors in this manuscript provide an approach of measuring the inequality in CO2 levels in 30 China provinces from 2005 to 2015. The manuscript falls well within the scope of Sustainability, and it deals with a very interesting and hot topic, that is the environmental pollution in China.
The manuscript is very well written and structured.
It consists of a well written and concise literature review, while the methods are described in detail.
The results are presented in a structured and reader friendly way, and they are properly and adequately discussed.
I would suggest moving the equations to an Appendix; the make the manuscript unnecessarily long, and they steal focus from the aims of this work.
I would suggest using navy blue and live red in Figures 1 and 2.
